# Knowledge Translation: A New Pathway for Model Compression

## Abstract

Deep learning has witnessed significant advancements in recent years at the cost of increasing training, inference, and model storage overhead. While existing model compression methods strive to reduce the number of model parameters while maintaining high accuracy, they inevitably necessitate the re-training of the compressed model or impose architectural constraints. To overcome these limitations, this paper presents a novel framework, termed **K**nowledge **T**ranslation (KT), wherein a "translation" model is trained to receive the parameters of a larger model and generate compressed parameters. The concept of KT draws inspiration from language translation, which effectively employs neural networks to convert different languages, maintaining identical meaning. Accordingly, we explore the potential of neural networks to convert models of disparate sizes, while preserving their functionality. We propose a comprehensive framework for KT, introduce data augmentation strategies to enhance model performance despite limited training data, and successfully demonstrate the feasibility of KT on the MNIST dataset. Code is available at the supplementary material.

## 1 Introduction

Deep learning constitutes a pivotal domain within computer science research. Despite the proposition of numerous innovative and practical models, which have made notable contributions to societal advancement, a significant proportion of these models heavily depend on increasingly intricate and larger architectures, as well as enhanced computational power, to elevate performance benchmarks (Schwartz et al., 2020). As a consequence, these large-scale models necessitate considerable resources for both training and inference, which leads to resource inefficiencies and impedes widespread application. Generally, two strategies can be contemplated to mitigate this problem. The first strategy is to design compact model architectures; nevertheless, this may not optimally leverage the existing large models that have already been trained. The second strategy, which is the focus of our paper, is to employ model compression methods.

Existing model compression methods (Choudhary et al., 2020) can be principally classified into four categories: low-rank factorization (Kishore Kumar & Schneider, 2017), pruning (Liang et al., 2021), quantization (Gholami et al., 2022), and knowledge distillation (Gou et al., 2021). However, these methods typically have certain drawbacks. As shown in Table 1, **knowledge distillation requires training the compressed model from scratch, while others impose limitations on its architecture**. Owing to the recurrent parameter updates of models in practical applications, it poses a significant challenge when the model compression algorithm requires full training. Moreover, investigating diverse combinations of modules aids in pushing the performance boundaries, as exemplified by Neural Architecture Search (Ren et al., 2021). Nonetheless, the inability to utilize inconsistent architectures hampers the potential for such advancements. Unfortunately, current model compression methods cannot simultaneously address both of these issues. As a result, the adoption of a superior architecture, such as Mamba (Gu & Dao, 2023), forces enterprises to abandon their existing Transformer (Vaswani et al., 2017) architecture and engage in the strenuous process of training a new model from scratch. This transition imposes substantial resource overhead as model compression techniques like pruning and quantization are rendered unviable due to the inconsistent architectures.

Table 1: Comparison of model compression methods.

| Method | Without re-training | Inconsistent architectures |
|---|:---:|:---:|
| Low-rank factorization | ✓ | ✗ |
| Pruning | ✓ | ✗ |
| Quantization | ✓ | ✗ |
| Knowledge distillation | ✗ | ✓ |
| Knowledge translation (ours) | ✓ | ✓ |

Therefore, we present a brand new approach to harness the potential of model compression, which "translates" the parameters of a large model into parameters suitable for a smaller model using a deep neural network. **This approach allows us to convert a large model into a different compact one without the need for re-training.** This idea is inspired by language translation, where distinct languages conveying identical meanings can be translated utilizing the deep learning models. As illustrated in Figure 1, building upon the success of language translation, our approach utilizes on the commonalities among disparate networks, which exhibit analogous functionalities, such as the extraction of texture and edge features from objects. Based on this insight, we pose the question of whether it is possible to achieve a "translation" between two distinct models (or modules). Since model parameters can be regarded as a kind of knowledge of how to realize their functionality, we name this approach **K**nowledge **T**ranslation (KT). Despite the simplicity of this idea, to the best of our knowledge, no prior attempts have been made. **Consequently, our work does not aim to achieve immediate and substantial performance triumphs. Instead, it takes the initial step towards validating the feasibility of KT, with the goal of capturing researchers' attention and fostering subsequent exploration within this domain.**

Our work makes the following key contributions:

- We provide a comprehensive explanation of the methodology used to train a knowledge translation model and successfully validate its feasibility on the MNIST dataset.

- We propose data augmentation methods to alleviate the challenges associated with training knowledge translation models when the amount of available training data is insufficient.

- We identify and discuss several potential research directions, thereby paving the way for further advancements in the field of knowledge translation.

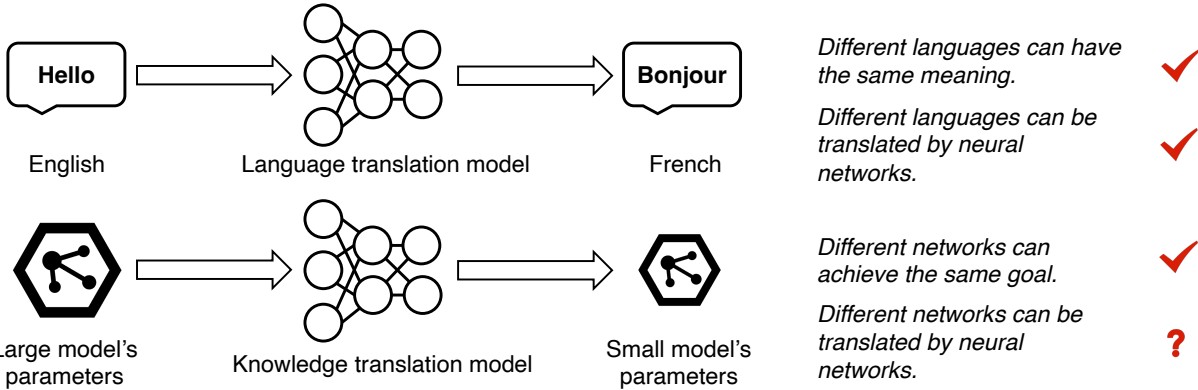

Figure 1: Schematic comparison of language and knowledge translation.

## 2 Related Work

### 2.1 Green AI

Green AI (Schwartz et al., 2020) refers to more environmentally friendly and inclusive AI research. Unfortunately, the dominance of scientific research today still revolves around obtaining higher accuracy through the utilization of larger models, more data, and elaborate experiments (Red AI). However, pushing the performance boundaries in many domains is becoming increasingly challenging, with the resource overhead growing at a much faster speed than the increase in accuracy (Schwartz et al., 2020). Fortunately, there is a positive trend emerging as more and more research attention is directed towards compact models. Notable works, such as LLaMA (Touvron et al., 2023) in natural language processing, Mirasol (Piergiovanni et al., 2023) in multimodal learning, and efforts to simplify the Transformer architecture (He & Hofmann, 2023), demonstrate the tremendous potential of Green AI.

### 2.2 Model Compression

Model compression is an important way to achieve Green AI (Choudhary et al., 2020) and it can be broadly categorized into four directions.

**Low-rank factorization.** Considering that both convolutional and fully connected layers can be seen as weight matrices, low-rank factorization (Kishore Kumar & Schneider, 2017) strives to replace them with matrices of smaller ranks to reduce parameter numbers. Various methods can be employed, including singular value decomposition (Stewart, 1993), tucker decomposition (Tucker, 1966), and canonical polyadic decomposition (Hong et al., 2020). However, factorization is computationally expensive, and the model architecture is somewhat limited.

**Pruning.** Given that neural networks often contain redundant parameters, the technique of pruning (Liang et al., 2021) aims to eliminate these redundancies and achieve efficient compression without significant performance compromise. Pruning can be performed with regard to weights (Han et al., 2015), neurons (Hu et al., 2016), filters (Li et al., 2016), and layers (Chen & Zhao, 2018).

**Quantization.** Quantization (Gholami et al., 2022) reduces computational requirements by decreasing the precision of parameters. One popular approach is mixed precision (Micikevicius et al., 2017), which is widely used for training large models. While numerous quantization methods have been proposed for efficient training (Wang et al., 2018; Banner et al., 2018) or inference (Wu et al., 2020; Yao et al., 2021), similar to low-rank factorization and pruning, the compressed architecture still faces certain limitations.

**Knowledge distillation.** Knowledge distillation (Gou et al., 2021) involves the use of a large pre-trained "teacher" model and a compact "student" model to be trained. By aligning their outputs, knowledge can be transferred and the student performance is improved. Initially, the knowledge is transferred in the form of logits (Hinton et al., 2015), but it later includes other forms such as layer-wise features (Romero et al., 2014; Chen et al., 2022; Sun et al., 2023) and cross-layer features (Chen et al., 2021). Knowledge distillation is not constrained by architectural requirements between the teacher and student models, but requires higher training overhead since the student model needs to be trained from scratch.

## 3 Knowledge Translation

In this section, we illustrate knowledge translation using an example of the image classification task. Given the complexity of translating the entire model, we focus on translating individual blocks instead. Specifically, we translate a large intermediate block within a network, into a smaller intermediate block. As shown in Figure 2, we categorize these steps into three main stages: generating input data (Step 1), generating target data (Step 2), and training the knowledge translation model (Step 3).

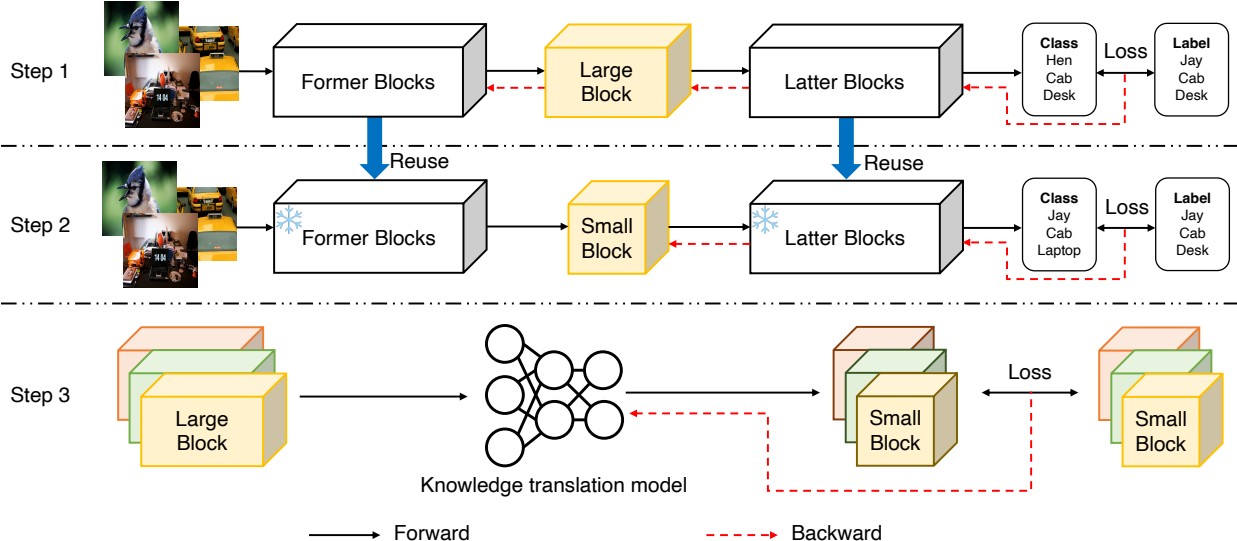

Figure 2: Overview of knowledge translation. Better viewed in color.

Briefly, we describe methods for generating training data in Section 3.1. Considering that data generation can be costly, in Section 3.2, we propose feasible data augmentation approaches to alleviate the problem. In Section 3.3, we elucidate the way to train knowledge translation models.

### 3.1 Data Generation

Data plays a crucial role in deep learning training, and it holds significant importance in the context of knowledge translation. Our first step is to acquire a sufficient amount of large model parameters to serve as input data for the knowledge translation model. In this process, we feed images into the model to obtain predicted results. We then calculate the loss by comparing them with the actual labels, and perform back-propagation to update the parameters of the entire model.

Once we have obtained the input data, the next step is to acquire the corresponding target data. **It is important to remember that the fundamental concept of knowledge translation is to achieve the transformation between network parameters of different sizes while preserving the functionality**. To ensure this, in this step, we need to keep network parameters fixed, except for the one being translated. As illustrated in Figure 2, during Step 2, we initialize the parameters of the former and latter blocks using the parameters obtained in Step 1. It is crucial to ensure that these parameters remain unchanged throughout the training process.

### 3.2 Data Augmentation

It is relatively easy to construct a dataset for language translation by collecting bilingual corpora from the Internet. However, when it comes to knowledge translation tasks, data collection becomes challenging. Since we need to get the trained parameters of large and small models as data, this means that getting a single piece of translation data requires two training processes. While training on datasets like MNIST is relatively fast, it is challenging when dealing with larger datasets.

While crop, flip, and rotation techniques are commonly employed for data augmentation in image classification tasks, these methods may not be suitable for knowledge translation tasks due to significant differences between the model parameters and the images. Hence, we propose two data augmentation methods that are well-suited for knowledge translation: random masking and noise addition.

**Random masking.** Random masking bears resemblance to dropout (Srivastava et al., 2014). Dropout is employed to discard certain neurons in order to improve generalization, while ensuring the network's ability to inference accurately. Similarly, since our data consists of model parameters, it is crucial to enable approximate inference even when some parameter values are set to zero. Building on this observation, for parameters of any size, we can randomly generate a mask of the same size, with all values uniformly distributed between 0 and 1. Using a hyper-parameter $m \in [0, 1]$, we can reassign the mask values accordingly. If the value at a particular position in the mask is less than $m$, it is reassigned to 0; otherwise, it is reassigned to 1. By element-wise multiplying the parameter with the mask, we generate an augmented data sample.

**Noise addition.** Noise addition is another effective approach, leveraging the inherent robustness of the model. Given that minor perturbations to the model's parameter values should not heavily impact its inference performance, we can introduce noise that obeys the normal distribution $N(0, 1)$ and has the same size as the parameter. By multiplying this noise by a hyper-parameter $n$ to control its range, we can subsequently add it to the parameter. This process generates new augmented data.

### 3.3 Model Training

By repeating steps 1 and 2 enough times, we end up with a set of corresponding knowledge translation datasets. In our work, we aim to compress the classical "BasicBlock" in ResNet (He et al., 2016) into a smaller version. As illustrated in Figure 3, in the small block, we remove the batch normalization layers and reduce the number of channels in the convolutional layers by half. While the large block utilizes batch normalization layers, we do not incorporate their parameters as input during the knowledge translation process.

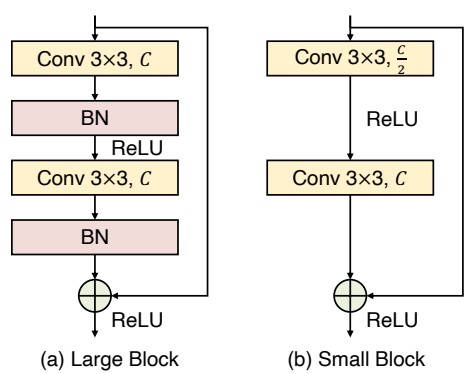

(a) Large Block    (b) Small Block

Figure 3: Example block architectures for knowledge translation.

Hence, we now have two weight matrices for the input convolutional layers and another two weight matrices for the output convolutional layers. The weight matrix of the 2D convolutional layers has a size of $(C_{\text{out}}, C_{\text{in}}, K, K)$, where $C$ represents the channel number and $K$ represents the kernel size. Consequently, the input can be represented as two tensors with dimensions $(B, C, C, 3, 3)$, and the output can be represented as one tensor with dimensions $(B, \frac{C}{2}, C, 3, 3)$ and another tensor with dimensions $(B, C, \frac{C}{2}, 3, 3)$, where $B$ is the batch size.

To train our translation model, we simply calculate the loss between the predicted and the target parameters, and then perform back-propagation. In this paper, we choose mean square error (MSE) as the translation loss, which is denoted as

$$L = \frac{\sum_{n=1}^{N} \text{MSE}(M_n^{\text{predicted}}, M_n^{\text{target}})}{N}, \tag{1}$$

where $M$ denotes the parameter matrices and $N$ is the number. The last question that needs to be answered is: what architecture should we choose for the translation model?

## 4 Pilot Experiment on Translation Model Architecture

To determine a suitable architecture for knowledge translation model, we evaluate the fitting ability using three commonly used architectures in deep learning networks: multi-layer perceptrons (MLP), attention, and convolution. This evaluation is performed on a small dataset consisting of 1,024 sets of corresponding parameters. We set the batchsize to 128 and train for 100 epochs, while remaining FLOPs (floating-point operations) approximately consistent across the different architectures. Additionally, we employ the same optimizer, learning rate, and weight decay value of 0 for all experiments.

### 4.1 Comparison Result

We utilize the MSE training loss to assess the fitting ability of different architectures. A lower training loss indicates a better fitting ability. Figure 4 demonstrates that the superiority of MLP architecture, while the attention architecture shows slower convergence to the training data. We propose two potential factors contributing to this phenomenon.

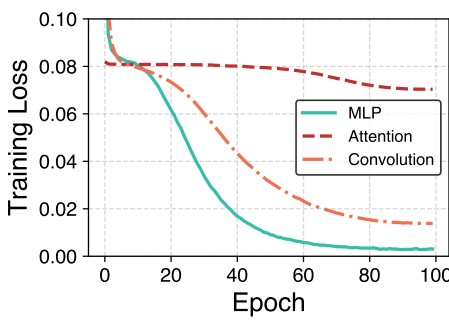

First, this phenomenon can be attributed to the network design. For instance, wide and shallow networks, despite having similar FLOPs, can yield significantly different outcomes compared to narrow and deep networks. However, the varying architectures require different designs, making it challenging to maintain consistency. Second, it could be influenced by the inherent architecture characteristics. For instance, the tensors within the attention are obtained through weighted combinations of previous tensors, which imposes constraints on their value range. In contrast, MLP does not encounter such limitations.

Figure 4: Fitting ability evaluation for different architectures.

### 4.2 MLP-Mixer

Considering the aforementioned reasons, we have opted to utilize a MLP architecture for the knowledge translation model. However, this does not imply that other types of architectures are not viable. We firmly believe that exploring novel architectures is crucial to advance our method.

We have selected MLP-Mixer (Tolstikhin et al., 2021), a novel variant of the MLP network. MLP-Mixer exclusively comprises normalization and activation layers, with the exception of the fully connected layer. Unlike traditional architectures, it does not incorporate convolution or attention structures. For the image classification task, MLP-Mixer accepts inputs of size $(B, D_S, D_C)$, where $D_S$ represents the sequence dimension and $D_C$ represents the channel dimension. By utilizing transposition operations, MLP-Mixer effectively integrates spatial information and per-location features, facilitating input dimension interaction and ultimately achieving outstanding performance.

Therefore, in the knowledge translation task, we reshape and concatenate the input two tensors into the shape of $(B, 2 \times C \times C, 3 \times 3)$ and feed it into the network. In the initial stage of the network, we employ fully connected layers to map the sequence and channel to higher dimensions. Towards the conclusion of the network, we employ fully connected layers to map both the sequence dimension and the channel dimension, while utilizing division and reshaping operations to achieve the desired tensor size.

## 5 Experiment

### 5.1 Setting

**Dataset.** We validate the feasibility of our proposed knowledge translation on MNIST (LeCun et al., 1998), which is a dataset comprising handwritten digits. It consists of 60,000 training images and 10,000 test images. MNIST encompasses 10 distinct classes that represent integer values ranging from 0 to 9.

**Training details.** We generate a total of 300,000 sets of corresponding parameters specifically designed for the MNIST classification task, which serve as our training data. We employ Adam optimizer with an initial learning rate of 0.001. Throughout training with 300 epochs, we utilize cosine annealing to dynamically adjust the learning rate, without incorporating weight decay, gradient clip, or warmup techniques. The batch size is set to 4,096. The channel number $C$ of the large block in Figure 3 is set to 6.

**Evaluation details.** We generate 100 sets of parameters for the large model (model in Figure 2 Step 1), comprising former blocks, large blocks, and latter blocks. The parameters from the former and latter blocks

Table 2: Details of the model architectures.

| Specification | **S**mall | **B**ase | **L**arge | **W**ide |
|---|---|---|---|---|
| Number of layers | 8 | 16 | 24 | 2 |
| Sequence length | 72 | 72 | 72 | 72 |
| Hidden size | 128 | 128 | 128 | 512 |
| MLP dimension $D_S$ | 256 | 256 | 256 | 256 |
| MLP dimension $D_C$ | 512 | 512 | 512 | 2048 |
| Parameters (M) | 1.32 | 2.61 | 3.90 | 4.11 |

are reused in the small model (model in Figure 2 Step 2), while the parameters from the large block are input into the trained knowledge translation model to obtain predicted small block parameters. These inferred small models are then used to calculate the accuracy on the MNIST test set. After every 25 training epochs, we compute the average test accuracy across the 100 models.

**Model architecture.** We conduct experiments on different model architectures, which are detailed in Table 2. The model architectures we explored include the **S**mall, **B**ase, and **L**arge models. These models have the same sequence length, hidden size, and MLP dimensions, but differ in the number of layers. Additionally, we investigate the **W**ide model, which has fewer layers but a wider hidden size and MLP dimension $D_C$. Despite these differences, the **W**ide model has similar parameters and FLOPs compared to the **L**arge model. In the subsequent experiments, we apply dropout (Srivastava et al., 2014) with varying probabilities $p \in \{0, 0.1, 0.2, 0.3\}$ to these models and report the results.

## 5.2 Compared Method

Given that our approach does not require complete re-training or constrained by model architectures, it may not be directly comparable to existing model compression methods. As such, we showcase our superiority by comparing with some specific parameter assignment techniques.

**Random initialization.** The parameters of the compressed module are randomly initialized.

**Random replacement.** We randomly select 100 sets of small block parameters from the training set to serve as evaluation parameters. These parameter values have a more reasonable magnitude and distribution.

**Greedy replacement.** To obtain the compressed parameters for each large block during evaluation, we compute the MSE distances between it and all the large blocks in training set. We then select the parameters of the small block associated with the training set's large block that exhibits the smallest distance.

## 5.3 Result

As shown in Table 3, our proposed knowledge translation achieves a significant accuracy enhancement for the compressed model. But beyond that, we anticipate that these findings will provide insights into a question that has intrigued us: **Are knowledge translation networks involved in computing, rote memorization, or learning?**

By excluding the batch normalization parameters in the large blocks during translation, we have discarded the complete input parameter information. Consequently, if the model is merely engaged in straightforward computation, it can easily lead to failures.

The poor performance of random replacement demonstrates that despite using the same training method and dataset, the inherent randomness in the training process still leads to a wide range of choices for the final model parameters. This observation is further supported by the low accuracy achieved through greedy replacement. The failure of this method implies that solely memorizing training data cannot achieve high

Table 3: Comparison of accuracy (%) with parameter assignment and knowledge translation using various model architectures. **Bold value** indicate the optimal result.

| Type | Model | Dropout | Best accuracy |
|---|---|---|---|
| Random initialization | | | 27.38 |
| Random replacement | - | - | 23.67 |
| Greedy replacement | | | 26.53 |
| | **S**mall | 0 | 45.54 |
| | | 0.1 | 31.08 |
| | | 0.2 | 30.23 |
| | | 0.3 | 30.63 |
| | **B**ase | 0 | **47.33** |
| | | 0.1 | 32.49 |
| | | 0.2 | 31.20 |
| | | 0.3 | 31.19 |
| Translation | **L**arge | 0 | 44.99 |
| | | 0.1 | 39.88 |
| | | 0.2 | 34.13 |
| | | 0.3 | 31.14 |
| | **W**ide | 0 | 30.31 |
| | | 0.1 | 30.98 |
| | | 0.2 | 31.63 |
| | | 0.3 | 29.08 |

evaluation accuracy, while our knowledge translation model effectively learns the intrinsic features of the data rather than merely memorizing them.

We visualize the features of some images in Figure 5. For the features obtained from the former blocks, additional feature extraction is performed on both the trained big and small blocks. This process results in a larger receptive field and less distinct boundaries for the number, as compared to the input feature. Since the parameters of the other blocks remain fixed during the training of the small block, similar features can be obtained for the large and small blocks.

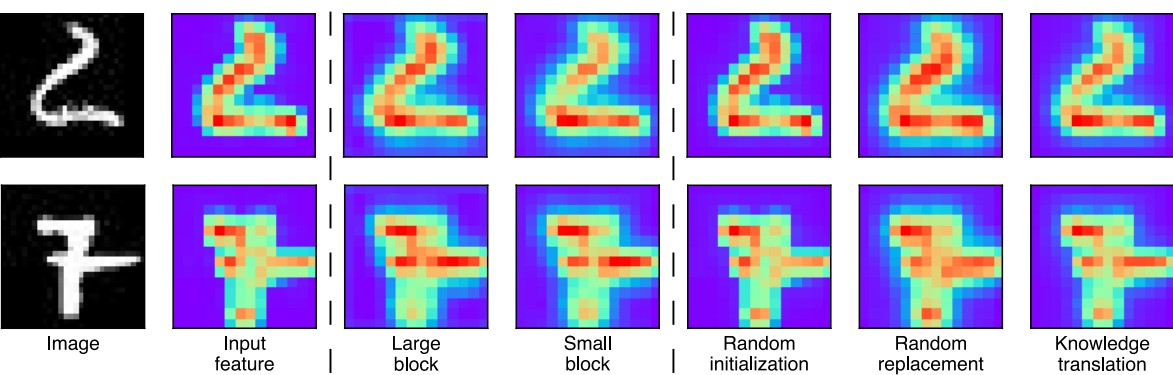

Figure 5: Visualization of features obtained using various methods. **B**ase model trained with 300 epochs is used for knowledge translation. Better viewed in color.

Table 4: Comparison of accuracy (%) and improvement (%) when using longer training epochs. **Bold values** indicate the optimal results. Improvement is the difference between the optimal accuracy of the current model and the optimal accuracy of the corresponding model presented in Table 3.

| Model | Dropout | Best accuracy | Improvement |
|---|---|---|---|
| **S**mall | 0 | 46.56 | +1.02 |
| | 0.1 | 39.85 | |
| | 0.2 | 31.87 | |
| | 0.3 | 31.66 | |
| **B**ase | 0 | 46.16 | +2.08 |
| | 0.1 | 49.41 | |
| | 0.2 | 47.11 | |
| | 0.3 | 32.77 | |
| **L**arge | 0 | 51.21 | **+7.08** |
| | 0.1 | **52.07** | |
| | 0.2 | 50.00 | |
| | 0.3 | 49.31 | |

As for random initialization, the model typically assigns parameters close to 0, which can limit the effectiveness of feature extraction. As a result, the output of the blocks heavily relies on the identity component, leading to features that closely resemble the input features.

On the other hand, random replacement aids in feature extraction but may cause the model to incorrectly focus on key locations, resulting in degraded performance.

In contrast, knowledge translation produces parameters that accurately extract features at the appropriate locations, making them more similar to the features obtained from the trained small block. Consequently, it yields improved performance.

Furthermore, despite having more parameters and higher FLOPs, the **W**ide and **L**arge models fail to surpass the performance of the **B**ase model. Regarding the **W**ide model, despite observing a substantial decrease in training loss during the training process, its accuracy consistently falls below 32% in all evaluations (Figure 6). We speculate that this wide but shallow architecture may not be well-suited for knowledge translation within our specific setting. We thus decide not to utilize it in the subsequent experiments.

Regarding the **L**arge model, we speculate that there are two potential reasons: First, its larger size requires more training iterations to achieve satisfactory results, but the current epoch numbers may be insufficient. Second, the **L**arge model may be more susceptible to over-fitting issues, which limits its overall performance.

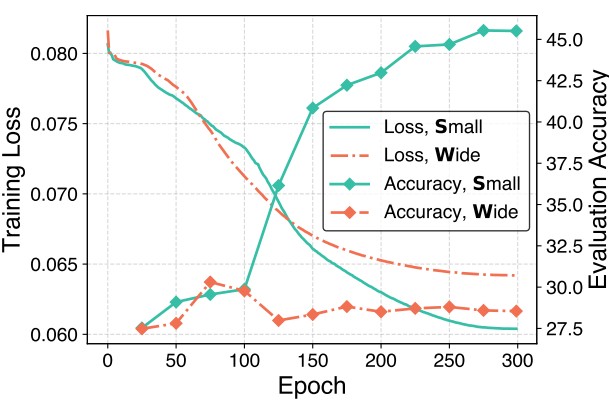

Figure 6: Training loss and evaluation accuracy (%) for **S**mall and **W**ide models.

**Longer training epochs.** Therefore, we increase the number of training epochs to 1,000. As shown in Table 4, this yields performance improvements for all models. Among them, the **L**arge model exhibits the most significant improvement and achieves the highest evaluation accuracy. We also notice that a higher dropout rate is more beneficial with 1,000 training epochs. These results confirm our previous hypotheses.

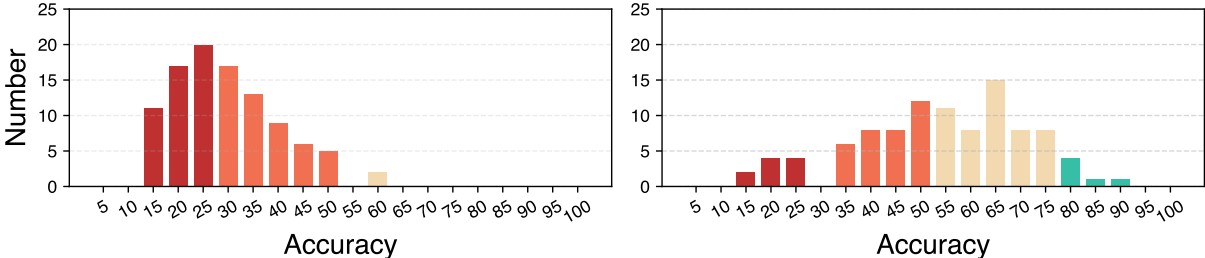

Figure 7: Evaluation accuracy distribution for random initialization (left) and knowledge translation (right). The accuracy (%) $a$ on the x-axis indicates that the evaluation accuracy falls within the range $[a - 5\%, a)$.

Table 5: Comparison of accuracy (%) with different augmentations. All results are obtained using the **L**arge model. **Bold value** indicate the optimal result.

| Dataset size | Training epochs | Augmentation | Best accuracy |
|---|---|---|---|
| 300,000 | 300 | - | 44.99 |
| 50,000 | 1,800 | - | 30.93 |
|  |  | Random masking | 48.76 |
|  |  | Noise addition | **49.52** |

**Accuracy analysis.** We also analyze evaluation accuracies using the **L**arge model trained for 1,000 epochs, comparing it to the distribution of evaluation accuracies obtained from random initialization. The results in the Figure 7 demonstrate that knowledge translation is not solely dependent on a specific number of high accuracy instances to boost evaluation metrics. Instead, it indicates that the model learns the underlying patterns within the data, rather than merely memorizing specific examples. Notably, knowledge translation effectively reduces the occurrence of low accuracies while increasing the likelihood of achieving high accuracies, resulting in an overall improvement in evaluation performance.

**Data augmentation.** A desirable expectation is that knowledge translation can still deliver satisfactory performance even with a reduced training dataset size. Unfortunately, as depicted in Table 5, when the dataset size is decreased while keeping the number of training samples fixed, the model's performance experiences a substantial decline. This observation underscores the necessity of having an ample amount of data for successful knowledge translation. As depicted in Table 5, both data augmentation methods have been proven to be effective in enhancing model generalization, addressing the issue of limited data, and significantly improving model performance.

## 5.4 Translation for Different Architectures

In this section, we demonstrate that knowledge translation is applicable to transformations between different architectures.

The results are presented in Table 6, revealing that knowledge translation leads to an improvement compared to random initialization. However, the magnitude of improvement is not as substantial as observed in Section 5.3.

We propose several reasons that may contribute to this outcome. Firstly, we notice a notable decline (>10%) in the upper performance limit of the model after replacing convolutions with other architectures. This decline could be attributed to our selection of MLP and attention architectures, which have a smaller parameter numbers. While these architectures are less computational efficient than convolution (Zhao et al., 2021), they may struggle to fully exploit the performance potential at a smaller parameter count. Moreover,

Table 6: Comparison of accuracy (%) when translating convolution to different architectures.

| Target architecture | Best accuracy | |
|---|---|---|
| | Random initialization | Knowledge translation |
| MLP | 26.60 | 37.33 |
| Attention | 26.72 | 33.55 |

the disparities between the architectures pose challenges for the knowledge translation model in learning their associations, ultimately impacting the quality of the translation performance.

Therefore, it is crucial to continue exploring and investigating different architectural choices, training techniques, and other pertinent factors for translation models.

### 5.5 Applicability

**Translating models with different training degrees.** Moreover, we explore a knowledge translation application scenario where a large enterprise trains a comprehensive translation model and offers model compression services to users. In this scenario, the enterprise possesses abundant training data and high-quality models, while the user has limited data available. As a result, the performance of the neural network used by the user for compression will likely differ from the performance of the neural networks employed by the enterprise to train the translation model. To evaluate the effectiveness of the translation model in this scenario, we firstly train the neural network parameters using the complete MNIST data for translation model training. Next, we employ this trained model to compress the parameters of neural networks trained with incomplete data. Finally, we compare the performance difference between knowledge translation and random initialization. As illustrated in the Figure 8, despite the variations in the data that lead to different models being compressed, knowledge translation remains effective in enhancing model performance across a significant range of differences.

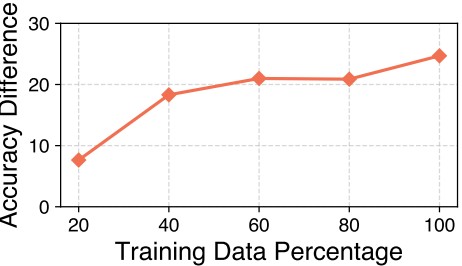

Figure 8: Accuracy difference (%) between knowledge translation and random initialization with different training data percentages (%). **L**arge model trained with 1,000 epochs is used. Evaluation is performed using 100 sets of model parameters for each percentage.

**Translating models from another dataset.** We also train 100 classification models on USPS (Hull, 1994) to assess the effectiveness of the translation model trained on MNIST in compressing them. Since the distribution of parameters obtained from training on different datasets can vary, we adopt a strategy of reusing and fixing the large block parameters from our validation set. We then train the remaining blocks to obtain the model that needs to be compressed. By using the pre-trained translation **L**arge model, knowledge translation leads to an average accuracy improvement of 11.11% compared to random initialization. It highlights the potential of transferring knowledge translation models to other datasets when the parameter distributions are similar. Furthermore, it showcases the practicality of efficiently training a unified translation model by incorporating parameters from different tasks and models, and offering compression services that cater to a diverse array of tasks and models.

## 6 Limitation and Future Work

While we have conducted preliminary investigations to validate the feasibility of knowledge translation on the MNIST dataset, we acknowledge that there are numerous pressing challenges that need to be addressed before the large-scale application of this approach, just as Rome was not built in a day. In light of this, we

put forward several potential research directions, aiming to harness the collective wisdom of researchers to propel the field forward.

**Architecture design.** In this paper, the utilized knowledge translation architecture is capable of handling inputs and outputs of fixed sizes. While certain models, like ResNet (He et al., 2016) and MLP-Mixer (Tolstikhin et al., 2021), exhibit repetitive architectural designs that enable the use of a limited number of knowledge translation models to process all modules and leverage branch networks for diverse outputs, the wide range of model parameters necessitates a well-designed knowledge translation architecture that can seamlessly handle various forms of network parameters. These may include MLP, convolution, attention, normalization, and so on. Additionally, we envision this architecture to provide a high degree of freedom in terms of output type and size options. A unified architecture also facilitates more efficient utilization of training data, enabling models to effectively learn the underlying fundamental patterns that are concealed within the parameters.

**Dataset construction acceleration.** As discussed in Section 3.2, having a sufficient amount of data is paramount for training knowledge translation models. Accelerating dataset construction becomes essential when dealing with larger datasets. Currently, in the construction of the training dataset, we only utilize the model parameters obtained at the end of training for knowledge translation. Exploring ways to fully leverage high-quality model parameters throughout the training process could be a promising avenue for future research.

**New data augmentation methods.** Data augmentation serves as a valuable technique to enhance model generalization without incurring additional training costs. Particularly in scenarios where generating training data is challenging, it offers an efficient means to improve model performance. Hence, utilizing the characteristics of deep learning networks to devise appropriate data augmentation methods, which enable model parameters to exhibit diversity while preserving their inherent functionality, is a promising direction for consideration and further exploration.

## 7 Conclusion

We introduce knowledge translation as a novel approach for achieving model compression. Traditional methods for model compression either necessitate re-training or lack flexibility in selecting the compressed architecture. In contrast, knowledge translation takes the parameters of a large model (or module) as input and utilizes a pre-trained "translation" model to "translate" them into a small model (or module), keeping the functionality approximately unchanged. This approach liberates the model from architectural constraints without requiring re-training.

We emphasize the significance of data and proposed data augmentation methods to address the issue of insufficient data. We investigate the architecture of the knowledge translation model and successfully validate its feasibility using the MNIST dataset. In comparison to randomly initializing small module parameters, employing knowledge translation to obtain small module parameters leads to significant performance improvements.

Furthermore, we put forth several future research directions, and we eagerly anticipate the involvement of more researchers to foster advancements in related fields and facilitate the development of efficient and sustainable Green AI.

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
