# OpenReview forum: "Knowledge Translation: A New Pathway for Model Compression"
_TMLR — Withdrawn by Authors_

### Review · Reviewer_kj9t · 2024-01-31

**Summary Of Contributions:**

The paper proposed a new idea called Knowledge Translation (KT). The goal of KT is to train a neural network that maps parameters from a large model to parameters of a small model while maintaining the performance. Author studied the feasibility of KT with MLP-Mixer, Attention and Convolution networks on MNIST.

**Audience:**

No

**Claims And Evidence:**

No

**Requested Changes:**

The author need to redesign the motivation, algorithm and experiments of KT. The author needs to consider how to make the KT algorithm more computationally feasible and propose a reasonable solution.

**Strengths And Weaknesses:**

Strengths:

1. The idea of KT is interesting. It will be beneficial if we can use a pretrained network to directly translates model parameters from a well-trained model to that of a new model (such as Mamba mentioned in the paper).

Weaknesses:
1. The KT method is not computationally practical. Training a KT network requires a large amount of parameter pairs from large / small models. As stated in Section 5.1, the author collected 300,000 pairs for training a KT network on MNIST. Neural networks that reach the quality of real-life applications has parameter size at least in the order of millions (e.g., the MobileNet series). Large-language models will easily go to the scale of billions or even trillions. According to the method described in Figure 2, collecting more than 300K pairs will require retraining these large / small networks for a tremendous amount of times, and also requires significant storage due to the large size of the parameter vectors.
2. The experimental design of the paper is not realistic. As mentioned in Section 5.1, ``the channel number C of the large block in Figure 3 is set to 6.``. The number of channels will easily go over the hundreds for neural networks. Due to the unrealistic hyper-parameter choice, the accuracy on MNIST is also very poor. Neural network can easily reach 90% accuracy in MNIST while the best performance the author reported is 52.07% in Table 4.
3. The author conducted experiments for translating parameters from different architectures in Section 5.4, but has not mentioned the experimental details.
4. The author claimed in "Section 2.2--Knowledge Distillation" that the student network needs to be retrained. This is not true as we can trim a large model to get the student, for example, select layers from a 12-layer BERT teacher model to obtain a 3-layer BERT student model.
5. In the current form, KT has the same limitation as knowledge distillation in that it is only applicable when the architecture of large / small models are predefined. Once the KT network is trained, we cannot apply it to translate parameters of other type of architectures.

---

### Review · Reviewer_hhhN · 2024-02-08

**Summary Of Contributions:**

The paper proposes a seemingly new approach for model compression, dubbed as knowledge translation. The ultimate goal of the proposed method is to design another machine (e.g. a trained NN) that gets parameters of a large NN and generates parameters for a smaller and compressed NN. The authors propose a general framework and test it on simple NN architectures on MNIST dataset. However, the method doesn't seem to perform quite well on MNIST and the successful adaptation of the method to more challenging cases is questionable.

**Audience:**

Yes

**Claims And Evidence:**

No

**Requested Changes:**

The work needs more rigorous evaluation of the idea, either theoretically or via simulations.

**Strengths And Weaknesses:**

*Strengths*:
 - The proposed venue of research is interesting, and if it is shown that such an idea can work, would be an interesting tool to be used for model compression.

*Weaknesses*:

The major weakness of the paper is that the authors fail to show that such an idea can actually work or extend to other scenarios. Their approach to solve the knowledge translation is to prepare a large dataset of the parameters of original large models and compressed ones and feed them to another NN to learn the mapping. Apart from the feasibility of creating such a dataset, the generalization of the trained NN (i.e., knowledge translator) to other datasets or different model architectures is largely doubted and not addressed.

Apart from this main issue, there are some other concerns such as

1- The authors claim that "knowledge distillation requires training the compressed model from scratch, while others impose limitations on its architecture." However, their own proposed method suffers from both drawbacks more seriously. The data gathering requires training more models and the knowledge translation NN is practically limited to the architectures that it has been trained on. The proposed method doesn't have a method to feed the architecture of original or desired compressed models.

2- The block-wise approach to the compression (i.e., fixing former and latter blocks) forces the input and output dimensions of the intermediate large block that we want to be compressed to be equal to the original model. This eliminates many compressed models that have fewer neurons or channels in their hidden layers.

3- In section 3.3, second paragraph, why there is batch size B in the number of parameters (or dimension) of input and output?

4- There are some concerns about the simulations. Although the paper seems to be a preliminary results exploring an idea, but basically most ideas work on simple datasets such as MNIST. The architecture of the compressed models and original models (i.e., inputs and outputs of knowledge translation) are not defined, and it is not clear if the trained model performs uniformly on different architectures. The results are also unsatisfactory. Knowing that accuracy of simple models on MNIST is above 97%, the reported results points that the method is not actually working in the simplest scenario. Moreover, the comparison should be to some other reasonable compression algorithm. Claiming that since this paper is exploring new venue for model compression doesn't justify that being better than random initialization is good enough.

5- The work doesn't have any theoretical analysis to justify the lack of rigorous experimental validation.

---

### Review · Reviewer_kiQ7 · 2024-03-25

**Summary Of Contributions:**

The paper presents a framework named "Knowledge Translation" (KT) for model compression. KT trains a "translation" model to convert the parameters of a larger model into compressed parameters, drawing inspiration from language translation techniques. The framework includes data augmentation strategies to improve performance with limited training data and demonstrates KT's feasibility on the MNIST dataset. Key contributions include the KT methodology and data augmentation methods to address training data limitations.

**Audience:**

Yes

**Broader Impact Concerns:**

1. **Resource usage**: The training of translation models could require significant computational resources. Discussing the environmental impact of increased energy consumption and how the proposed method compares to traditional model compression techniques would be useful.


2. **Bias and fairness**: Any compression technique might inadvertently amplify or introduce biases present in the original model. The authors can address how the KT approach might affect model biases and what measures can be taken to mitigate this risk.

**Claims And Evidence:**

Yes

**Requested Changes:**

#### Critical changes:
1. **Expand experimentation**: Test the Knowledge Translation approach on more complex datasets beyond MNIST to demonstrate scalability and effectiveness on challenging problems.

2. **Efficiency analysis**: Provide a detailed comparison of the training time and computational resources required for the Knowledge Translation model versus traditional model compression methods.

3. **Comparative analysis**: Include a broader comparative analysis with existing compression techniques, especially those that do not require re-training from scratch.

#### Changes that would strengthen the work:
1. **Real-world applications**: Discuss potential real-world applications and practicality of the KT approach, including how it might be deployed in different architectural settings and domains.

2. **Architecture versatility**: Investigate and report the KT approach's performance across different neural network architectures.

3. **Parameter efficiency analysis**: Analyze the parameter efficiency of the translated models compared to their original counterparts and other compressed models.

**Strengths And Weaknesses:**

### Strengths

1. The paper treats model compression as a translation problem, which is interesting and creative.
2. The framework for Knowledge Translation is detailed, covering data generation, augmentation, and model training.
3. The paper suggests methods to generate more training data when there's not enough.


### Weaknesses

1. The testing is only done on the MNIST dataset, which is extremely simple and does not show how the approach works on real problems.
2. The approach requires training a new translation model, which may not be more efficient than some existing compression methods that don't need re-training.
3. There's a lack of comparison with a wide range of existing compression methods, especially those that also avoid re-training.
4. The paper does not discuss how practical this approach is for real-world use, including how it would work on different model architectures or tasks.

---

### Note · Authors · 2024-03-29

**Comment:**

Dear Editor and Reviewers,

After thorough consideration and evaluation, we have made the decision to withdraw the manuscript.

We deeply appreciate the time and effort you have devoted to evaluating our work. Your invaluable feedback and insights are immensely beneficial to us. We will further improve our work based on the feedback received.

**Withdrawal Confirmation:**

I have read and agree with the venue's withdrawal policy on behalf of myself and my co-authors.